# Infer global, predict local: Quantity-relevance trade-off in protein fitness predictions from sequence data

**Lorenzo Posani**◉¤*◔, **Francesca Rizzato**, **Rémi Monasson**◉◔, **Simona Cocco**◉*◔

Laboratory of Physics of the Ecole Normale Supérieure, CNRS UMR8023 & PSL Research, Sorbonne Université, Paris, France

◔ These authors contributed equally to this work.
¤ Current address: Center for Theoretical Neuroscience, Columbia University, New York, United States of America
* lorenzo.posani@gmail.com (LP); simona.cocco@phys.ens.fr (SC)

## Abstract

Predicting the effects of mutations on protein function is an important issue in evolutionary biology and biomedical applications. Computational approaches, ranging from graphical models to deep-learning architectures, can capture the statistical properties of sequence data and predict the outcome of high-throughput mutagenesis experiments probing the fitness landscape around some wild-type protein. However, how the complexity of the models and the characteristics of the data combine to determine the predictive performance remains unclear. Here, based on a theoretical analysis of the prediction error, we propose descriptors of the sequence data, characterizing their quantity and relevance relative to the model. Our theoretical framework identifies a trade-off between these two quantities, and determines the optimal subset of data for the prediction task, showing that simple models can outperform complex ones when inferred from adequately-selected sequences. We also show how repeated subsampling of the sequence data is informative about how much epistasis in the fitness landscape is not captured by the computational model. Our approach is illustrated on several protein families, as well as on *in silico* solvable protein models.

## Author summary

Is more data always better? Or should one prefer fewer data, but of higher relevance to the task to be performed? Here, we investigate this question in the context of the prediction of fitness effects resulting from mutations to a wild-type protein. We show, based on theory and data analysis, that simple models trained on a small subset of carefully chosen sequence data can perform better than complex ones trained on all available data. Furthermore, we explain how comparing the simple local models obtained with different subsets of training data reveals how much of the epistatic interactions shaping the fitness landscape are left unmodeled.

**Data Availability Statement:** Data and code used for this manuscript can be found in the following GitHub repository: https://github.com/lposani/fitness-prediction-tradeoff.

**Funding:** This work was supported by Agence Nationale de la Recherche (RBMPro CE30-0021-01 and ANR-19 Decrypted CE30-0021-01 to S.C. and R.M.). The funders had no role in study design, data collection and analysis, decision to publish, or preparation of the manuscript.

**Competing interests:** This work has no competing interests.

## Introduction

Predictability of evolution of organisms in fitness landscape has been a driving concept in the development of evolutionary biology since the origins of the field [1–5]. In particular, our capability to predict the effects of detrimental mutations has enormous practical impact on the diagnosis of genetic variances causing diseases [6–10]. This issue can now be quantitatively investigated, thanks to high-throughput sequencing and mutagenesis experiments, which allow for in-vivo and in-vitro measurements of the effects of many mutants [1, 5, 11–14, 14–25]. However, despite the impressive progress of these large-scale techniques, the number of possible mutations, growing exponentially with the protein length, is so huge that measuring the fitness landscape in its entirety is out of reach, with the exception of short protein regions [1]. Computational approaches, in particular machine-learning-based models exploiting the large corpus of available sequence data [26, 27] are needed for the full reconstruction and prediction of fitness landscapes. Briefly speaking, these methods are based on the assumption that statistically rare mutations (in homologous sequence data) are likely to be deleterious [6, 28]. Such conservation-based methods can be combined with structural [7, 29], physico-chemical [8], as well as phylogenetic [30, 31] information.

Graphical Potts models, also called direct coupling analysis (DCA) [32–34], have pushed further the approaches based on sequence conservation by including statistical couplings capturing pairwise amino-acid covariation. These couplings allow DCA to account for background effects on the mutations depending on the wild-type (*wt*) sequence under consideration. DCA is thought to approximate the fitness landscape reflecting the structural and functional properties common to homologous proteins. As sketched in Fig 1, natural sequences are assumed to lie at, or close to the different peaks of the fitness landscape explored during evolution. The scores of sequences around the *wt* protein provide predictions for the effects of single or multiple mutations on fitness, in good agreement with mutational effects measured through mutagenesis experiments [34–38]. Other approaches for fitness prediction exploit deep learning (DL) architectures, at the origin of recent progress in image or natural language processing, as well as in protein folding [9, 10, 39–41]. DL models have much higher expressive power than pairwise graphical models, but demand massive sequence data to be trained. Recent applications of DL to protein fitness modelling combine unsupervised learning of hundreds of millions of sequences with supervised learning of mutagenesis experimental data [38, 42, 43].

Depending on the protein family under consideration, multi-sequence alignments (MSA) show huge variations in sizes, with tens to hundred of thousands sequences, and in homology, ranging from ∼30%, for alignments of orthologous sequences [27, 37, 44], to 90%, for HIV sequences of the same clade [35, 45]. The quantity and diversity of the data, as well as the models considered are empirically known to strongly impact the performances for fitness prediction. As pointed out in [46], classical methods based on homology detection, such as SIFT [6], PolyPhen-2 [7], Align-GVGD [8], rely on different empirical procedures in selecting the alignments, and are not always optimal. Remarkably, single mutations effects are predicted with comparable accuracy by graphical models inferred from a small number of highly similar sequences of the HIV envelope protein [35] and from a much larger number of diverse sequences of Betalactamases, while the two proteins have comparable lengths [37]. Gemme, a recently introduced algorithm based only on conservation and phylogenetic tracing of mutations [31] was shown to outperform deep neural networks models [39] in predicting the effect of mutations in viral sequences, all characterized by a large degree of similarity. Furthermore, the performance of models trained from Uniprot sequences with high pairwise alignment score to a fixed *wt* sequence considerably vary with the threshold used for alignment [37, 46].

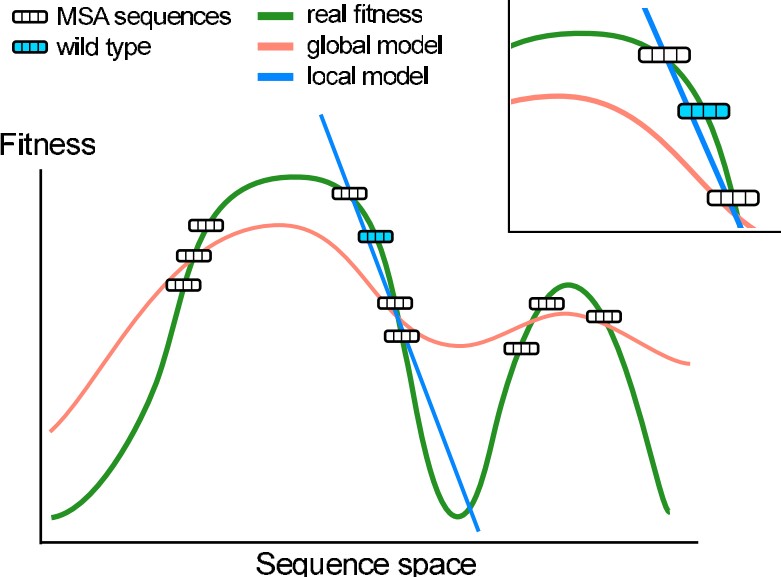

**Fig 1.** Schematic visualization of the fitness landscape over the sequence space (green curve). Two models (red and blue curves) are inferred to assign high fitness values to sequences found in the Multi-Sequence Alignment (MSA) of a protein family. A complex model (red curve) can be a better predictor of the landscape globally while scoring poorly in predicting single-point mutations around a specific wild-type sequence, see local fitness landscape in the zoomed area. Conversely, a simple model (blue line) fitted on a local subset of sequences can give a better local approximation of the landscape, but will likely fail in distant regions of the fitness landscape.

These examples suggest the existence of a compromise between taking into account many sequence data to get statistics and removing far away sequences, whose relation to fitness may be very different from *wt* due to complex epistatic effects. This compromise, in turn, depends on the expression power of the model considered, which can be tuned at will, and on the complexity of the fitness landscape, which is generally unknown. As sketched in Fig 1, on the one hand, predicting mutations around the *wt* requires *local* reconstruction of the landscape only, a task within reach of simple models with few defining parameters. These models are however unreliable for sequences far away from the *wt* sequence; hence, only few data points, concentrated around the latter can be actually used for training. On the other hand, powerful models able to capture the complex features, such as high-order epistasis that characterize the *global* fitness landscape on large scales can, in principle, exploit at best sequence data. However, even if the available data are sufficient to infer their huge number of parameters with enough accuracy, it is unclear whether the global description they offer allows for an accurate local reconstruction of the fitness landscape around the *wt* protein. The scope of the present work is to provide theoretical foundation to address this question.

Careful analysis of the different contributions to the prediction error allows us to quantitatively understand how fitness prediction performance depend on both model complexity and on the sequence data, and to estimate the amount of 'complexity' in the fitness landscape that is not captured by the model. Our theory is in full agreement with the analysis of sequence data and mutagenesis experiments for 7 protein families we have studied. We also validate our approach *in silico* on Lattice-Protein models [47–50], for which the ground truth for the fitness is mathematically well defined. Last of all, we demonstrate how our framework allows us, in practice, to optimally tune sequence alignments and models to maximize the performance in fitness prediction.

## Results

### Quantity-relevance trade-off in MSA sequence selection

We consider a reference sequence, hereafter referred to as *wt*. We denote by $\mathcal{E}_{ia}$ the variation of fitness resulting from the mutation $wt_i \rightarrow a$ on the $i^{th}$ site of *wt*. This quantity can be estimated experimentally, either in vivo (relative enrichment of organisms with mutated gene compared to *wt*), or in vitro (measurement of appropriate biochemical property).

A computational model provides a predictor, $\hat{\mathcal{E}}_{ia}$, for the difference of fitness between the mutant and the *wt*. The overall quality of the predictor will be assessed through the Spearman coefficient $\rho$ between the mutation effects computed with the model $\hat{\mathcal{E}}_{ia}$ and with the experimental data $\mathcal{E}_{ia}$. Using Spearman correlations allows one to capture monotonous relations, irrespective of non-linearities.

The computational model is generally trained from homologous sequences to *wt*, *i.e.* belonging to the same protein family. The similarities between the *wt* and these sequences, sampled from evolutionary diverse organisms, can vary significantly. As an illustration, we consider the RNA binding domain of the nuclear poly(A)-binding protein (PABPN1), involved in the synthesis of the mRNA poly(A) tails in eukaryotes [14]. Any two sequences in the corresponding MSA (as used in [37]) generally have few amino acids in common (mean Hamming distance -normalized by sequence length- between pairs of sequences in the MSA = 0.75). As a result, a specific sequence, such as the *wt* of *Saccharomyces cerevisiae*, is generally surrounded by a small number of similar sequences and is far away from most of the MSA (RNA-bind protein: mean normalized Hamming distance between *wt* and MSA sequences = 0.73, Fig 2A; see S1 Fig for similar results on other families).

Hereafter, we show that sequences far away from *wt* are not relevant for fitness prediction. To do so we train independent-site Potts models (Methods) on shorter MSAs obtained by discarding sequences further than a certain cut-off distance $d_{cut}$ from *wt*. As $d_{cut}$ becomes smaller, fewer sequences with higher proximity are selected (Fig 2A). We see that the performance consistently increases when decreasing the cut-off distance, up to a peak $\rho = 0.56$ at $d_{cut} = 43$, a 33% increase with respect to the full MSA ($\rho(d_{cut} = 82) = 0.42$), see Fig 2B and 2C. After peaking, the performance starts decreasing again due to the increasingly-lower number of sequences in the MSA, see Fig 2C.

The non-monotonous behavior of the predictive performance indicates that a trade-off between the number of sequences and their proximity to *wt* is controlling the predictive performance of the inferred model. To investigate the respective effects of these two quantities, we create sub-alignments of the original MSA with controlled sizes $B$ (effective number of sequence taking into account sequence redundancy, see [51] and Methods) and average Hamming distances to *wt*, which we denote as $D$. We then test how the performance of the independent-site Potts model trained on these sub-alignments relates to these two quantities.

This analysis showed that the predictive performance strongly depends on the mean Hamming distance $D$ and on the number $B$ of sequences ($P < 0.001$ for all Spearman correlations between $\rho$ and $B$ or $D$, Fig 2D). The performance significantly decreases with $D$ at fixed $B$, *i.e.*, when the relevance of the data deteriorates and their quantity is kept fixed, and increases with $B$ at fixed $D$, *i.e.*, when the quantity of data increases at fixed similarity with *wt*. Similar results are found for the six other protein families under study (see S1 Fig).

### Theoretical investigation of the quantity-relevance trade-off

To study the trade-off between relevance and quantity we draw our inspiration from the bias-variance framework developed in statistics [52, 53]. Let us consider the error $\hat{\mathcal{E}}_{ia} - \mathcal{E}_{ia}$ between

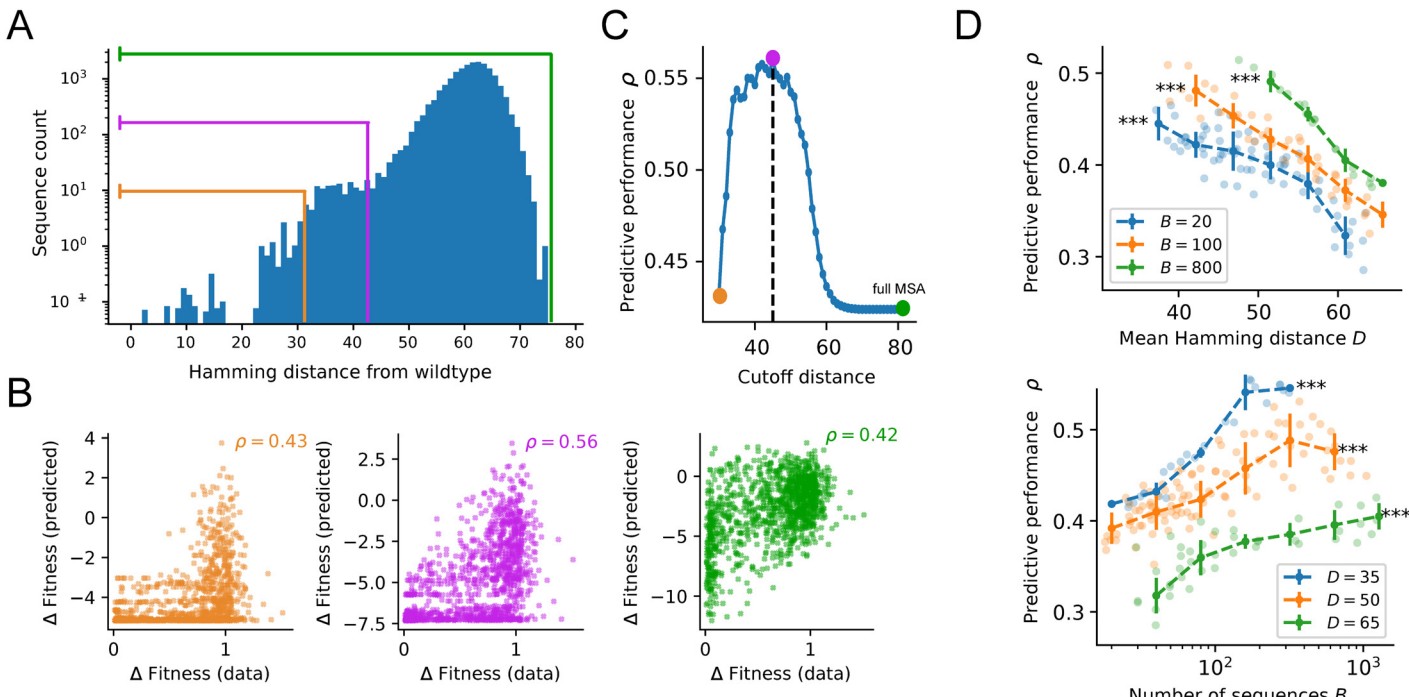

**Fig 2. Behaviour of model predictive performance with different selections of training data. A**. Distribution of Hamming distances to the *wt* sequence (RNA-binding domain of Pab1-Yeast) in the MSA of [37]. Note the log scale on the *y* axis. The three colored lines correspond to three possible sequence selections performed by excluding sequences farther than a certain threshold $d_{cut}$ from *wt*. A smaller $d_{cut}$ corresponds to fewer sequences with a lower mean Hamming distance to the *wt*, denoted as *D*. **B** Comparison between predicted and experimental fitness mutational effects for an independent-site model trained on the three sub-MSAs corresponding to, respectively, $d_{cut} = 32$ (orange), 43 (purple), and 82 (green). The Spearman correlation coefficient *ρ* between predicted and experimental values defines the predictive performance of the model. **C** Same analysis as panel B repeated for all possible cutoffs between $d_{cut} = 32$ and $d_{cut} = 82$ (the sequence length). The non monotonous behavior of the predictive performance indicates that a trade-off between number of sequences (denoted as *B*) and proximity to *wt* is controlling the predictive performance of the inferred model. **D**. Systematic analysis of the predictive power *ρ* as a function of the mean Hamming distance *D* of sub-alignments with fixed size *B* (top), and of the sub-alignment size *B* at fixed Hamming distance *D* (bottom). Each individual point shows the average over *n* = 5 sub-samples obtained at the corresponding values of *D* and *B* (see Methods). The dashed curves and error bars are computed by binned average and standard deviation over the displayed individual points. All significance levels refer to Spearman rank correlation of the individual points. *** $P < 0.001$.

the statistical predictor $\hat{\mathcal{E}}_{ia}$ and the experimental fitness $\mathcal{E}_{ia}$. This error can be decomposed into the sum of two contributions: (1) a systematic bias in the prediction, due to the inability of the model to capture the exact relation between sequence mutation and fitness, (2) a statistical error coming from the fact that the predictive model has been trained on a particular data set; the value of this contribution fluctuates when the data set changes, and is expected to be smaller and smaller for larger and larger data sets.

Consequently, the mean squared error on the single-point mutation $wt_i \to a$ can be written as the sum of a *squared bias* and a *variance* contributions,

$$\underbrace{\left[(\hat{\mathcal{E}}_{ia} - \mathcal{E}_{ia})^2\right]}_{\text{mean squared error}} = \underbrace{([\hat{\mathcal{E}}_{ia}] - \mathcal{E}_{ia})^2}_{\text{squared bias } \mu_{ia}^2} + \underbrace{\left[(\hat{\mathcal{E}}_{ia} - [\hat{\mathcal{E}}_{ia}])^2\right]}_{\text{variance } \sigma_{ia}^2}, \tag{1}$$

where averages [] are taken on repetitions of the prediction process in fixed conditions (relevance and quantity of data). Notably, these two quantities are hard to minimize together. For instance, powerful models with many parameters will accurately fit the data and thus achieve small squared biases $\mu_{ia}^2$, but will result in large variances $\sigma_{ia}^2$ due to the statistical errors on the many parameters to be inferred. As we will see below, we can directly relate our descriptors of

relevance ($D$) and quantity ($B$) of the sequence data to, respectively, the squared bias and the variance as defined in (1). Furthermore, we will introduce a class of increasingly powerful Potts models to investigate the effect of model complexity on these two quantities. In addition to its theoretical appeal and close connection with the bias-variance decomposition, considering the mean–squared error is ultimately justified by the empirically observed monotonic relation with the predictive performance as measured through the Spearman coefficient $\rho$.

**K–link Potts model.**   We consider hereafter the class of sparse Potts models, which include $K$ pairwise couplings between the protein sites, $J_{ij}(a, b)$, whose values depend on the amino acids they carry and a field (position weight matrix) $h_i(a)$ on each site; These parameters are learned from the MSA (Methods). The choice of the $K$ pairs of sites carrying couplings is decided based on heuristics, which aim at capturing interrelations between the residues (Methods).

By tuning the value of $K$, we can interpolate between the independent-site model ($K = 0$, *i.e.* no coupling) and the full Potts model ($K = K_{max} \equiv \frac{1}{2}N(N − 1)$ couplings, where $N$ is the protein length). Imposing small values of $K$ is a way to regularize the inferred network of interactions. Notice that the number of parameters to be inferred, $N_{par} = NQ + KQ^2$, where $Q = 20$ is the number of amino acids, grows quickly with $K$ since $Q^2 = 400$.

For the $K$-link Potts model the predictor of the fitness difference resulting from the mutation $wt_i \rightarrow a$ reads

$$\hat{\mathcal{E}}_{ia} = h_i(a) - h_i(wt_i) + \sum_{j \in \mathcal{N}_i} \left( J_{ij}(a, wt_j) - J_{ij}(wt_i, wt_j) \right) \quad , \tag{2}$$

where the sums runs on the sites $j$ in the neighborhood $\mathcal{N}_i$ of site $i$, *i.e.* coupled to $i$ (Methods). This neighborhood is empty for the independent-site model.

**Estimation of variance.**   For the Potts model, expressions for the uncertainties on the inferred fields $h_i(a)$ and couplings $J_{ij}(a, b)$ can be formally derived from sampling errors due to the finite size of the data set. The resulting variance $\sigma_{ia}^2$ of the predictor $\hat{\mathcal{E}}_{ia}$ for a specific K-link model can then be estimated from (2) [38, 54], see Appendix A in S1 Text. Averaging $\sigma_{ia}^2$ over the sites $i$ and mutations $a$, we obtain a single global variance,

$$\sigma^2 \simeq \frac{1}{B}\frac{1}{NQ} \sum_{i,a(\neq wt_i)} \left\{ \frac{|k_i - 1|}{p_i(a)} + \frac{|k_i - 1|}{p_i(wt_i)} \right. $$
$$\left. + \sum_{j \in \mathcal{N}_i} \left( \frac{1}{p_{ij}(a, wt_j)} + \frac{1}{p_{ij}(wt_i, wt_j)} \right) \right\} \quad , \tag{3}$$

where $Q = 20$ is the number of amino-acid types, and $k_i$ is the cardinality of $\mathcal{N}_i$, i.e. the number of sites interacting with $i$ in the model. The global variance depends on the statistics of the data through the probabilities $p_i(a)$ of finding amino acid $a$ on the $i$-th site and $p_{ij}(a, b)$ of finding simultaneously $a$ on site $i$ and $b$ on site $j$ computed on the sub-alignment. Thus, $\sigma^2$ increases with residue conservation, due to the contributions of amino acids that are rarely observed on some sites in the sub-alignment and have low $p_i(a)$, and with the number $K$ of coupling parameters in the model. We also see that $\sigma^2$ is inversely proportional to the number of sequences, $B$. The variance therefore decreases with the *quantity* of data.

**Estimation of squared bias.**   Computing the squared bias $\mu^2$ in (1) is generally hard, not to say impossible, as it requires detailed knowledge of the fitness landscape. We rely below on simplifying assumptions to gather insights on the value and meaning of the bias.

Assume first that we use the independent-site model for fitness prediction. If the 'true' fitness landscape shows no epistasis, this model is exact (up to statistical fluctuations due to the finite amount of training data, taken care of by $\sigma^2$), and the bias vanishes. Therefore, a non zero bias would signal the presence of epistatic interactions between residues not captured by the simple model used for predictions. We stress that this statement is true in an idealized setting, in which the only source of bias is the mismatch between the model power and the ground-truth fitness landscape. In reality, biases can have multiple origins, including non-uniform sampling of sequence data (resulting from preferential choices of organisms or from evolutionary correlations), discrepancies between in vivo fitness reflected by sequence data and in vitro biochemical measurements, etc.

Let us now turn to more complex landscapes and models. We assume that the fitness landscape is characterized by pairwise epistasis only, *i.e.* the fitness differences $\mathcal{E}_{ia}$ are exactly described by a full Potts model with $K_{max}$ interactions $J_{ij}^F(a, b)$ through an equation analogous to (2). The $K$–link Potts model used for fitness prediction will not be powerful enough to account for the complexity of this landscape and of the sequence data if $K < K_{max}$. As a result a non-zero squared bias will appear, whose expression is derived in Appendix B in S1 Text, and reads

$$\mu^2 \simeq J_0\, D\;, \tag{4}$$

where $D$ is the mean Hamming distance of the sub-alignment sequences to *wt*, and the *bias factor* $J_0$ is the product of a multiplicative factor depending on the background distribution of amino acids in the MSA and of the variance of the epistatic couplings $J^F$ *not included in the prediction model*. $J_0$ is thus a decreasing function of $K$.

This expression of $\mu^2$ confirms that the Hamming distance $D$ is related to the notion of *relevance* (similarity to the *wt*) of the sequence data, as varying $D$ affects the systematic error (bias) of the predictive model.

## Validation of the theory on Lattice Proteins

To validate the key role of the squared bias and of the variance in explaining performance, as well as their approximate expressions above and the interpretation of the bias factor $J_0$ as reflecting un-modeled epistasis, we resort to an *in silico* model for proteins folding on a 27-site cubic lattice [47, 49, 50, 55, 56], see Fig 3A. In the model, the fitness represents the propensity of a protein sequence to fold into one specific conformation, called native, out of the $\simeq 10^5$ folds on the cube [49]. Following [50], the native fold and wildtype sequence were chosen such that the fitness of the wildtype was high enough to be stable but low enough to allow for positive mutations ($P_{nat} \simeq 0.995$, see Methods). As we can precisely compute the exact value of the fitness, the ground-truth values of the squared bias and of the variance defined in (1) can be computed with great accuracy (see Methods); we hereafter denote these ground truth values by $\bar{\mu}^2$ and $\bar{\sigma}^2$.

**Bias and variance are sufficient to explain model performance.** Eq (1) stipulates that the mean squared error over fitness prediction depends on the sum of squared bias and variance of the fitness predictors. If the performance $\rho$ is, in turn, controlled by this mean squared error, we expect a relation such as

$$\rho = F(\mu^2 + \sigma^2)\;, \tag{5}$$

where $F$ is a decreasing function of its argument.

To test the validity of (5), we compare the values of $\rho$ obtained with the independent-site Potts models ($K = 0$) and different K-link Potts models ($K = 4, 8, 16, 24$) trained from various

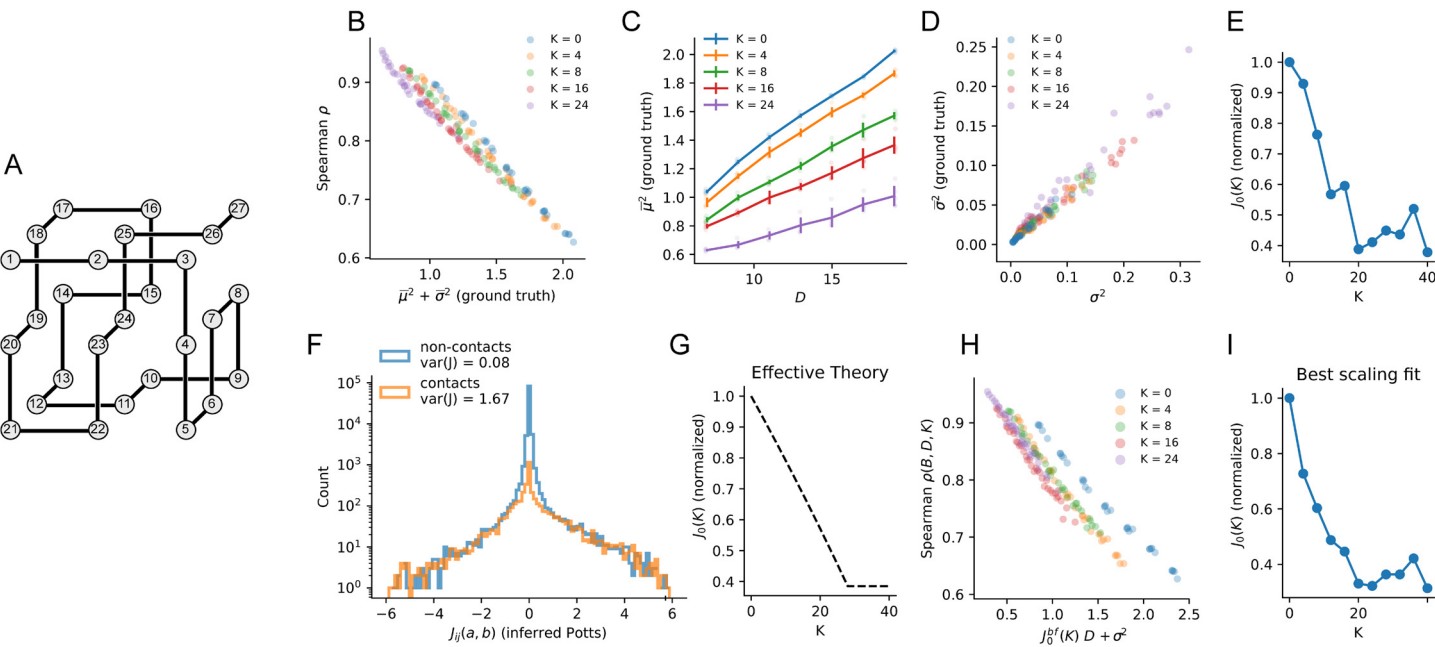

**Fig 3. Quantity-relevance trade-off for lattice proteins. A**: Cubic fold that defines the protein family in the lattice model. Amino acids on sites that are in proximity to each other interact and define the energy of the protein (Methods). **B**: Predictive performance $\rho$ for single mutations of 5 Sparse Potts models with different degrees of sparsity (defined by $K$, the number of pairwise links included in the energy function; $K = 0$ is the independent model) vs. $\bar{\mu}^2 + \bar{\sigma}^2$. The collapse of the results is in agreement with Eq (5). **C**: Squared bias $\bar{\mu}^2$ vs. mean Hamming distance in the sequence data, see Eq (4), for the same sparse Potts models as in panel B. Line plots and error bars show mean and standard deviation at a given $D$ and different $B$s. **D**: Variance $\sigma^2$ vs. estimated variance $\hat{\sigma}^2$ in Eq (3) for the same Sparse Potts models as in panel B. **E**: Bias factor $J_0(K)$ (divided by $J_0(0)$) obtained by fitting the squared bias as a linear function of the mean Hamming distance for the various K-link models in panel C. **F**: Visualization of pairwise couplings inferred by a fully-connected Potts model, highlighting the larger variance of couplings associated to structural contacts (in orange) compared to non-structural ones (in blue)—note the log scale on the y-axis. **G**: Normalized value of $J_0(K)$ (divided by $J_0(0)$) obtained with an effective theory using the variance of couplings associated to modeled and un-modeled structural contacts, see Appendix A in S1 Text. **H**: scaling for predictive performance $\rho$ of our statistical models for single point mutations as a function of the sum of the estimated squared bias $J_0 D$ and of the variance $\sigma^2$ in Eq (3). $J_0(K)$ (denoted as $J_0^{bf}$ in the plot axis label) is fitted to for each value of $K$ by maximizing the scaling correlation as explained in the main text. **I**: Bias factor $J_0(K)$ (normalized by $J_0(0)$) inferred from maximizing the scaling correlation as in panel H.

sub-alignments with different $B$, $D$ to the sums of the squared bias and variance, see Fig 3B. We obtain an excellent anti-correlation between $\rho$ and $\bar{\mu}^2 + \bar{\sigma}^2$ across a large range of values of $B$ and $D$, in full agreement with (5) ($R \sim 1$ for every $K$–link model). The sum of squared bias and variance is by far the biggest factor in determining the predicting performance of the models.

**Bias and variance are related to the relevance and the quantity of data as predicted by theory.** We then test the relation between the squared bias and the Hamming distance in (4), by generating MSAs at a given $D$ and numerically computing $\bar{\mu}^2$ for several K-link Potts model of increasing complexity. As shown in Fig 3C, the linear relation between the true squared bias and $D$ is confirmed for every value of $K$ ($R \simeq 1$ for every tested K-link model).

Similarly, we find a good agreement between the numerical variance and our theoretical estimate in (3), see Fig 3D ($R \simeq 1$ for every K-link Potts model).

**$J_0$ reflects the un-modeled epistasis.** The slope of the numerical bias $\mu^2$ with $D$ (Fig 3B) gives access to an estimate for $J_0$. We plot in Fig 3F the corresponding $J_0$ as a function of the number $K$ of links in the Potts model, from $K = 0$ (independent model) to $K = 40$. We find that $J_0(K)$ decreases almost linearly with $K$ before reaching a saturation point around $K = 20$.

This decrease is in accordance with the notion of $J_0$ as reflecting the un-modeled epistasis. In the context of Lattice Proteins, this saturation behavior is expected to reflect the presence of

two distinct classes of un-modeled epistatic couplings. Strong pairwise interactions correspond to the $N_c$ = 28 contacts on the 3D fold (Fig 3A). These "structural" couplings are expected to be largely responsible for the magnitude of epistatic effects in the fitness function, therefore contributing the most to the value of $J_0$. The remaining $K_{max} - N_c$ are weaker, and may be due to the need to avoid other folds (negative design) or to higher-order interactions [50].

To verify this hypothesis, we retrieve a pairwise approximation of the real fitness function by inferring a fully-connected Potts model from a very large alignment ($B \sim 10^6$ sequences). We then separate the inferred Potts couplings into structural and non-structural and compute their variance as a proxy for their expected contribution to the value of $J_0$ (see Appendix A in S1 Text). As shown in Fig 3F, structural couplings have a much larger variance than the other ones. We can devise an effective theoretical approximation of the behavior for $J_0(K)$ by assuming that all structural and non-structural couplings are uniformly drawn from two distributions with the two variances above, and that the sparse model progressively includes structural couplings in its energy function up to $K = N_c$. The expected behavior of $J_0(K)$ under this effective model, shown in Fig 3G, agrees with Fig 3E, and saturates to its lowest value around $K = 28$, which corresponds to the total number of structural couplings.

**$J_0$ can be inferred from mutational scan data.**   Last, we propose an alternative approach to estimate the bias factor $J_0$, which is applicable to real protein data, where the sequence-to-fitness mapping is unknown but mutational scans are available. For fixed model complexity (value of $K$), we subsample the MSA, infer the corresponding K-link Potts models, and estimate the predictive performances $\rho$. The procedure is repeated by varying the quantity ($B$) and relevance ($D$) of the sub-MSAs. We then consider $J_0$ as a free parameter and infer its value by maximizing the Spearman correlation between the two sides of (5), where $\sigma^2$ is estimated from Eq (3) and $\mu^2 = J_0 D$. We call this approach the "best scaling fit".

We apply this procedure to the same lattice protein data shown above. Results for the performance $\rho$ vs. $J_0 D + \sigma^2$ are shown in Fig 3H for all K-link Potts models ($R \simeq 1$ for every tested K-link model), in excellent agreement with the ground truth results of Fig 3B. The fitted values of $J_0(K)$ are reported in Fig 3I, in excellent agreement with Fig 3E and 3G.

## Performance vs. quantity and relevance of sequence data for real proteins

**Trade-off explains the predictive performance in mutagenesis experiments.**   The relation in (1), which we verified on *in-silico* proteins, postulates that the performance $\rho$ of the predictive model is controlled by the sum of the squared bias $J_0 D$, as an inverse proxy for the *relevance* of the sequence data, and of the variance $\sigma^2$, which inversely depends on the *quantity* of data. To test our theory on real data, we consider 7 different mutagenesis experiments on 7 proteins. For each protein, we sub-sample the corresponding MSA as done in Fig 2, to obtain sub-MSAs with a large range of values of $D$ and $B$, from which we can compute the estimated variance $\sigma^2$. We then compute the two descriptors $D$ and $\sigma^2$ from each sub-MSA, and compare them with the predictive performance inferred from the data.

As reported in Fig 4A, $D$ is a fairly good predictor for the performance of an independent-site Potts model (RNA-binding domain—absolute value of Spearman correlation coefficient $r_S$ between $D$ and $\rho = 0.70$), while the variance alone correlates more weakly with the predictive performance (RNA-binding domain—absolute value of Spearman correlation coefficient $r_S$ between $\sigma^2$ and $\rho = 0.25$). However, when the performance is compared to the sum of the squared bias and the variance, $J_0 D + \sigma^2$, the correlation can be made much higher through fitting of $J_0$ (RNA-binding domain—absolute value of Spearman correlation coefficient $r_S$ between $J_0 D + \sigma^2$ and $\rho = 0.95$, Fig 4B). This strong correlation is confirmed for the 7 protein

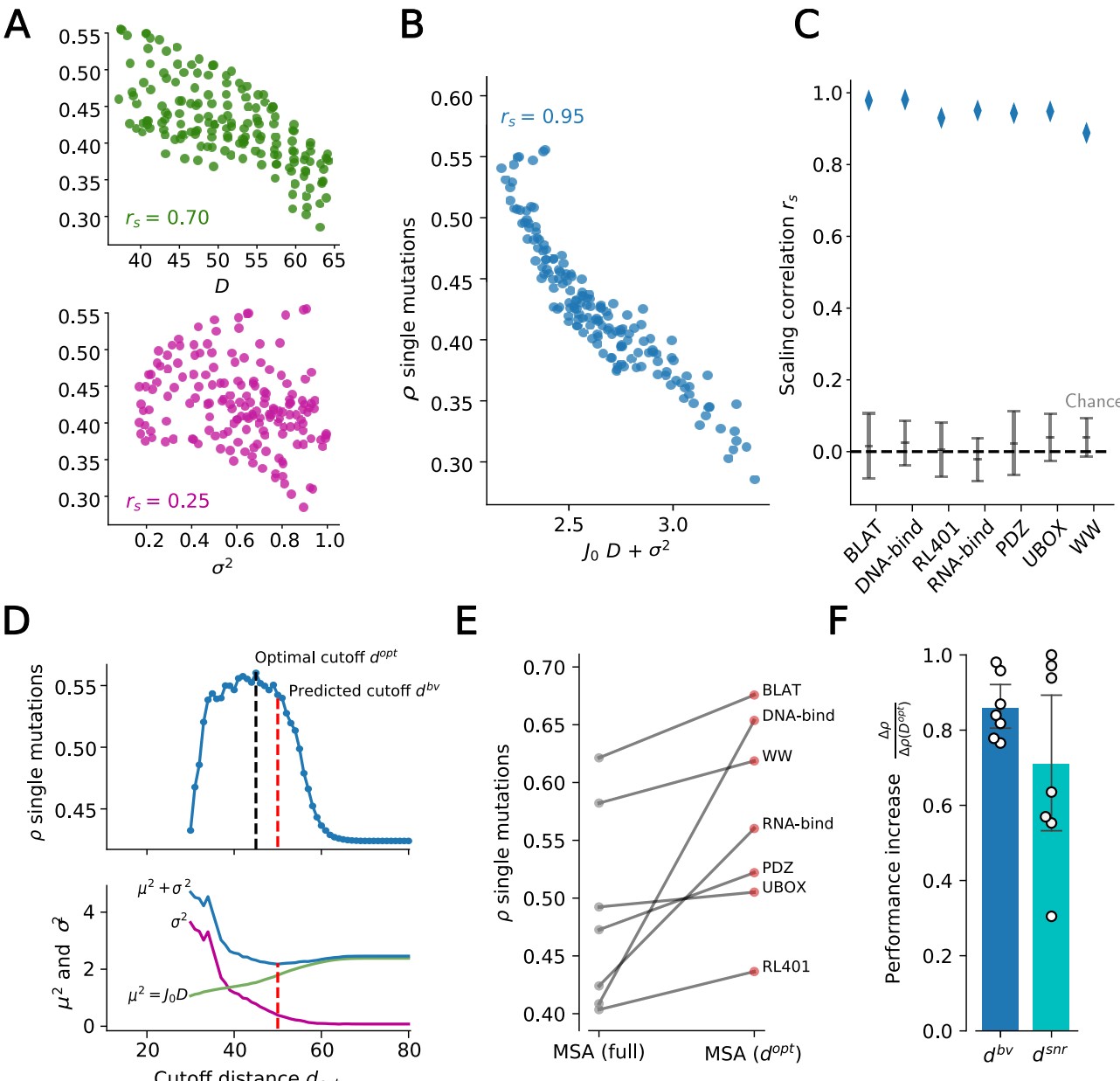

**Fig 4. Relevance-quantity trade-off explains the predictive performance of statistical modelling. A** predictive performance of single-point mutations using the Independent-site on the RNA-bind protein, shown as a function of the mean Hamming distance of the MSA (top) and variance estimated from the alignments (bottom). **B** predictive performance of single-point mutations as a function of the linear sum of squared bias and variance. The scaling correlation $r_S$ is computed as the absolute value of the Spearman correlation coefficient of $J_0D + \sigma^2$ vs. $\rho$. The bias factor $J_0$ is inferred by maximizing $r_S$, as done in [Fig 2E]. **C** scaling correlation $r_S$ for the seven protein families, compared to chance levels. The chance distribution is built by destroying the relationship between the performance $\rho$ and the two descriptors by random order shuffling, then repeating the $J_0$ inference procedure to account for the scaling optimization during its estimation. Error bars show standard deviations over $n = 100$ repetitions of the random shuffling. **D** top: RNA-bind family, predictive performance $\rho$ as a function of the cutoff distance $d_{cut}$, showing the existence of an optimal cutoff $d^{opt}$ (black dashed line). Bottom: individual contributions of squared bias ($J_0 D$, purple line), variance ($\sigma^2$, green line) and their sum (blue line). The red dashed line indicates the minimum of $J_0D + \sigma^2$, which corresponds to the predicted maximum performance cutoff $d^{bv}$. **E** Values of predictive performance $\rho$ at the optimal cutoffs compared to the full alignments for the 7 protein families. **F** ratio between performance increase at cutoffs of interest and at the optimal cutoff for the 7 protein families.

families ($r_S > 0.9$ for all 7 families, Fig 4C and S2 Fig), providing a strong verification of the theoretical and numerical framework developed above.

**Optimization of performance through a *focusing* procedure.** We may now exploit our understanding of how performance depends on the number $B$ and on the mean Hamming distance $D$ of the sequences in the MSA to find the optimal sub-alignments maximizing $\rho$.

As we see in Fig 2, we can start from the full MSA and progressively *focus* around *wt* by excluding all sequences of "low relevance", i.e., at Hamming distances higher than a given cutoff $d_{cut}$. As we lower $d_{cut}$ from its maximal value ($N$, number of sites) down to 0, this focusing procedure increases the variance while decreasing the bias, as we select fewer sequences with higher homology to *wt*. As already seen in Fig 2C, the predictive performance $\rho$ has a maximum at a certain optimal cutoff $d^{opt}$ (Fig 4D (top panel)), highlighting the trade-off between bias and variance in controlling the performance.

In Fig 4E, we report the performance of the independent-site model at the optimal cutoff $d^{opt}$. We find notable improvements in the predictive performance for 6 out of 7 protein families with respect to the full MSA (mean improvement $\Delta\rho(d^{opt}) = 0.081$). Importantly, for 3 families out of 7 (DNA-bind, RL401, WW), the value of $\rho$ at the optimal cutoff exceeded the best performance reported in [37] and obtained with PLM-DCA, a standard approach to learn the Potts model parameters [57]. This result is striking, as both the number of parameters and the number of training sequences involved in the inference at $d^{opt}$ are greatly reduced compared to fully-connected Potts models on large MSAs. The most outstanding illustration is the DNA-bind family, where top performance ($\Delta\rho = 0.26$) is found for $d^{opt} = 29$, corresponding to only $B = 37$ effective sequences in the MSA (see S3 Fig).

**Cutoff for optimal focusing can be reliably predicted from heuristics.** According to (5) the best performance is reached for the alignment that minimizes the sum $J_0 D + \sigma^2$. We call this optimal predicted cutoff $d^{bv}$, as for bias-variance,

$$d^{bv} = \operatorname{argmin} \{J_0\, D(d_{cut}) + \sigma^2(d_{cut})\} \tag{6}$$

As reported in Fig 4D(top) (red line) and Fig 4F, this procedure allows us to predict the optimal cutoff with good precision (mean relative error = 0.08). Importantly, the performance increase at the predicted cutoff $d^{bv}$ captures most of the total possible improvement (mean guessed relative increase for the 7 families $\Delta\rho(d^{opt})/\Delta\rho(d^{opt}) = 0.86 \pm 0.08$, see Fig 4F). Globally, the performance at the predicted cutoff $d^{opt}$ is systematically higher than the performance with the full MSA (mean $\Delta\rho(d^{opt}) = 0.073$, paired Wilcoxon test over the $n = 7$ families: $P = 0.018$).

However, knowledge of the bias factor $J_0$ entering Eq (6) is not always available, as it requires a systematic analysis of predictive performance relying on the outcome of mutagenesis experiments as a reference. We propose below a simple heuristics for predicting the optimal cutoff, requiring no experimental input and based on a signal-to-noise ratio (SNR) comparing the spread of inferred fitness values across sites and mutations and the statistical variance $\sigma^2$, Fig 4D(bottom):

$$\text{SNR}(d_{cut}) = \sqrt{\frac{\left[\hat{\mathcal{E}}_{ia}(d_{cut})^2\right]_{ia} - \left[\hat{\mathcal{E}}_{ia}(d_{cut})\right]_{ia}^2}{\sigma^2(d_{cut})}} \tag{7}$$

Setting for instance the cutoff $d^{snr}$ corresponding to a threshold of SNR = 3, we again find systematic improvements in the predictive performance (mean guessed relative increase for the 7 families $\Delta\rho(d^{snr})/\Delta\rho(d^{opt}) = 0.71 \pm 0.10$, see Fig 4F, S3 and S4(b) Figs), providing an unsupervised, parameter-free criterion to select the optimal MSA for the predictive analysis. Notice

that the choice of the value SNR = 3 above is arbitrary; A consistent improvement of performance can be found for SNR in the range $\sim 2$ to $\sim 4$, see S4(a) Fig.

**The bias factor $J_0$ depends on the model expressivity.** We repeat in Fig 5A the approach of Fig 4B, using the $K$–link Potts model rather than the independent-site model for fitness predictions. The number of couplings, $K$, is chosen to be a fraction of $N$, and is much smaller than $K_{max}$, implying that the Potts model is very sparse. For each sub-alignment of the RNA-binding domain data we determine the best scaling fit bias $J_0(K)$. We observe very high correlations between $\rho$ and $J_0(K)D + \sigma^2$. We also observe that top performances are found for a non-zero value of $K$, e.g. $K = 0.1N$ in Fig 5A. The optimal value of $K$ generally varies from family to family, as reported below.

The value of the bias factor $J_0(K)$ is shown as a function of the number of links per site in Fig 5B for the RNA-binding domain and for all 7 protein families in Fig 5C. The general behaviour is similar to the one observed for lattice proteins (Fig 3), and shows that $J_0(K)$

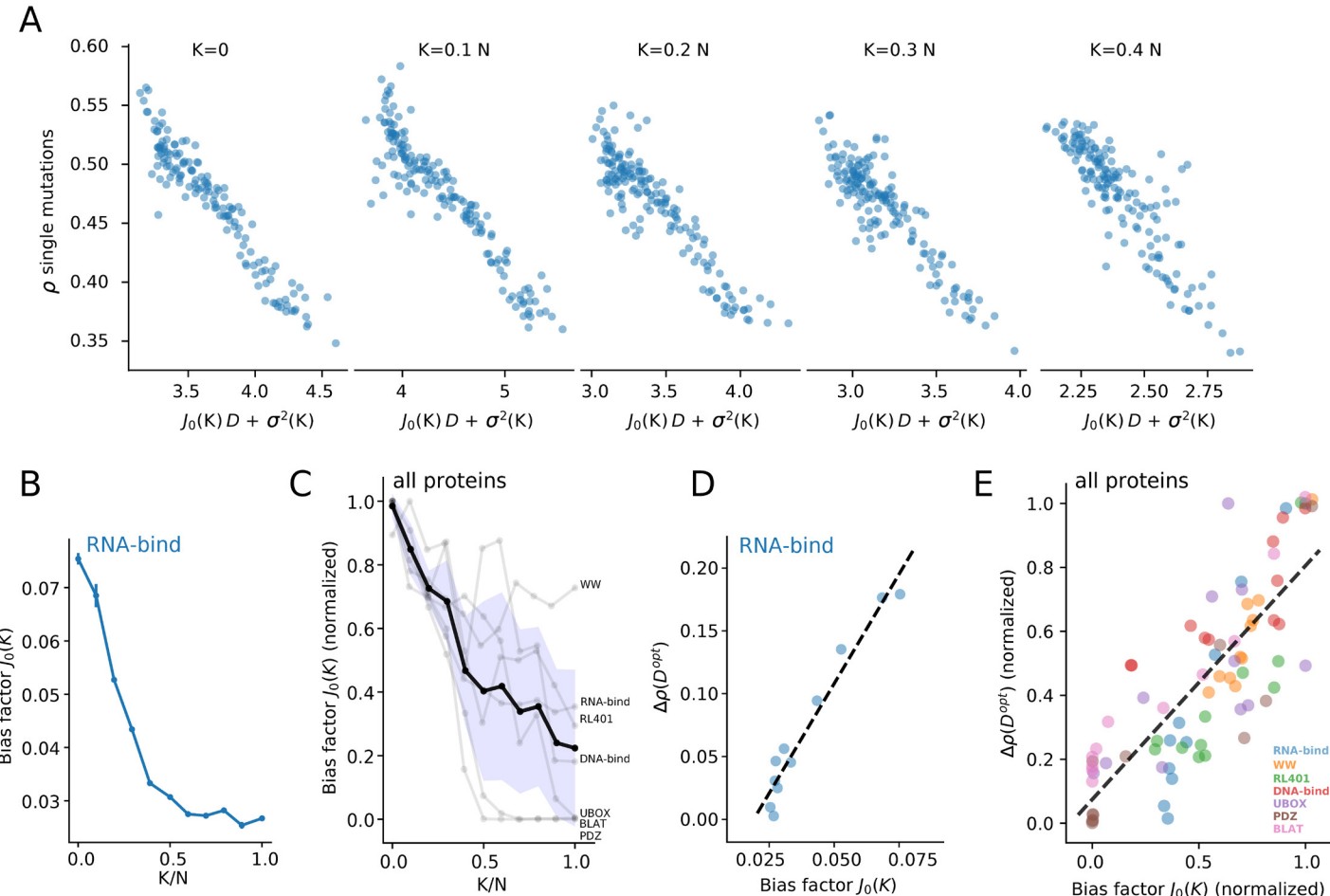

**Fig 5. The bias factor $J_0$ depends on the model expressivity. A** Scaling correlation between predictive performance $\rho$ and $J_0D + \sigma^2$ for the RNA-bind protein, modeled with the Sparse Potts Model with different numbers $K$ of couplings. $N$ is the length of the protein (82 sites). **B**: values of the bias factor $J_0$ as a function of the number of modelled couplings in the Sparse Potts Model for the RNA-bind protein. **C**: same as B for the seven protein families combined; the black line and the blue area represent the mean and the standard deviation over the seven protein families. **D** Relation between bias factor $J_0(K)$ and improvement at best cutoff $\Delta\rho(d^{opt})$ for the RNA-bind protein. **E** same of D for the seven families combined. Values of $K$ range from $K = 0$ to $K = N$. Each color corresponds to a different protein family as reported in the legend.

decreases with $K$ until saturation is reached. As the expressive power of the predictive model increases, the squared bias decreases and is less affected by the relevance of the sequence data. The saturation indicates that, above some critical $K$, adding more pairwise couplings does not help to reduce the bias. A possible explanation for this residual bias is the presence of higher-order epistasis, e.g. 3-site couplings between residues, which cannot be accounted for by the $K$–link Potts model.

Empirically, we expect that the focusing procedure should provide substantial improvement if the bias strongly decreases with $D$, that is, if the bias factor $J_0$ is large, *e.g.* in the case of the independent-site model. The intuition is that, when the bias quickly decrease with the relevance of the data, there is a margin for improvement of performances by removing some low-relevance data, while not increasing too much the statistical variance of the inferred model parameters. We report in Fig 5D the gain in performance $\rho$ (compared to the independent Potts model, with $K = 0$) for the RNA-binding domain as a function of the bias factor $J_0$ when $K$ is varied. Results show a strong positive correlation between the two quantities. The same correlation is found across all 7 protein families, see Fig 5E and S5 Fig.

## Materials and methods

### Multiple sequence alignments

Proteins families and the corresponding alignments were taken from [37]. The alignment procedure of EVmutation (https://github.com/debbiemarkslab/EVmutation) is based on a query against the UniRef100 database of nonredundant protein sequences (release 11/2015) [58] from the wild-type sequence, using the profile HMM homology search tool jackhmmer [59] and choosing a default score in the alignment depth of about 0.5 bits/residue; the threshold was adjusted if the alignment had not enough coverage or number of sequences [37]. Redundant sequences were removed from the alignments, as well as sites with more than 50% of gaps in the alignment. The list of families and wild type, the sequence length, the number of sequences, and the reference to the mutational scans are given in Table 1. Alignments are made available in Supplementary Information.

### Sequence re-weighting and MSA descriptors

We partially corrected for sampling biases by using a re-weighting procedure with 80% homology threshold in all statistical estimates on sequence data [32]. We therefore define the weight of a sequence $s$ to be

$$z(s) = \frac{1}{\sum_{s' \in MSA} \delta(d(s,s') < 0.2)} \ , \tag{8}$$

**Table 1. From left to right: Numbers $N$ of sites, $B$ of sequences (after removal of redundant sequences from the alignment), $M^1_{test}$ of tested single mutations, $M^1$ of possible single mutations, and corresponding references.**

| Protein Name (*wt* name) | $N$ | $B$ | $M^1_{test}$ | $M^1$ | Ref |
|---|---|---|---|---|---|
| Betalactamase (BLAT-ECOLX) | 253 | 7620 | 4610 | 4807 | [19] |
| DNA Binding domain (GAL4-YEAST) | 63 | 11278 | 1196 | 1197 | [60] |
| Ubiquitin (RL401_YEAST) | 71 | 10023 | 1160 | 1349 | [22] |
| RNA Binding (PABP-YEAST) | 82 | 70779 | 1188 | 1558 | [14] |
| PDZ (DLG4-RAT) | 84 | 24795 | 1577 | 1596 | [13] |
| UBOX Domain (UBE48-MOUSE) | 76 | 6153 | 614 | 1444 | [20] |
| WW (YAP1-HUMAN) | 31 | 8251 | 319 | 589 | [21] |

where $\delta(X) = 1$ if condition $X$ is satisfied, and 0 otherwise, and $d(s, s')$ is the normalized Hamming distance between sequences $s$ and $s'$. The MSA descriptors were then computed as weighted averages:

$$B = \sum_{s \in MSA} z(s) \quad \text{and} \quad D = \frac{1}{B} \sum_{s \in MSA} z(s)\, d(wt, s) \; . \tag{9}$$

## Fitness predictions and comparison with experiments

With the $K$-sparse Potts model the probability of the sequence $s$ reads

$$P(s) \propto \exp\left( \sum_i h_i(s_i) + \frac{1}{2} \sum_i \sum_{j \in \mathcal{N}_i} J_{ij}(s_i, s_j) \right) \tag{10}$$

up to a normalization constant. $\mathcal{N}_i$ denotes the set of sites connected to site $i$ on the interaction graph. The predicted fitness difference is defined as the difference in the log probabilities of the wild-type sequence ($wt$) and of the one where the amino acid at site $i$ in is substituted with the amino acid $a$ as $wt_{i \to a}$:

$$\hat{\mathcal{E}}_{ia} = \log P(wt_{i \to a}) - \log P(wt) \; , \tag{11}$$

giving Eq (2). The predictive performance $\rho$ of the model is then computed as the Spearman rank correlation between experimental measures of delta-fitness for single-point mutations and the corresponding predictions.

## Inference of sparse Potts models

Following a number of recent works [34, 36–38, 61], we predicted the effects of single point mutations by inferring a Potts model from sequence data in the alignment. Here, we employed a K-link Potts model introduced in [38], where we constrain the model to have non-zero couplings only on $K$ statistically-relevant links $(i, j)$ ($K = 0$ being the independent model, $K = N(N - 1)/2$ the fully connected Potts model).

We chose the $K$ links by scoring each link $(i, j)$ as done for contact prediction in DCA analysis: we inferred a fully-connected Potts model with parameters optimized to perform contact prediction by pseudo-likelihood maximization [57]; from the resulting couplings $\mathbf{J}^{PLM}$ we defined a score for each link $(i, j)$ based on the Frobenius norm of the two-sites coupling matrix $\mathbf{J}^{PLM}_{ij}$. Finally, we selected those $K$ pairs $(i, j)$ that showed the highest Frobenius score. We then used a two-site approximation to re-infer the value of the $\mathbf{J}_{ij}$ matrix for each of these $K$ pairs given the sparsity constraint [38].

## Sub-sampling the MSA allows for varying data relevance and quantity

To create new MSAs of different degrees of relevance and quantity for real protein families, we sub-sampled the corresponding MSA using the following procedure. We first chose a target Hamming distance $D$ and a number of sequences $B_0$ (before re-weighting). We then randomly sampled $B_0$ sequences $s$ from the full MSA (without repetition) with a probability decreasing with the Hamming distances $d(s, wt)$ between the sequences $s$ and the wildtype:

$$p(s) = \frac{e^{-\alpha\, d(s, wt)}}{Z} \tag{12}$$

where $Z = \sum_{s' \in MSA} e^{-\alpha d(s', wt)}$. The parameter $\alpha$ was optimized to reach the defined $D$ within a given precision (here set to 0.01).

From each sampled sub-MSA, we then computed the effective number of sequences $B$ as described above, as well as the variance $\sigma^2$ for each sparse Potts model with $K$ links through (3).

We repeated this procedure for all combinations of 16 values of $D \in [0.4N, 0.8N]$, where $N$ is the protein length, and 10 values of $B_0$ in a range that depended on the protein family and its initial MSA size. Doing so, we obtained a population of 160 sub-MSAs with as many corresponding values of $D$ and $B$.

## Lattice proteins

Lattice proteins are *in silico* proteins of fixed sequence length ($N = 27$) folding on the sites of a $3 \times 3 \times 3$ cube [47, 49, 55]. The protein *family* attached to a specific fold $F$ is defined as the set of sequences $s$ with low (favorable) folding energies $\epsilon(F, s)$ in $F$ and unfavorable folding energies $\epsilon(F', s)$ for all other possible folding structures $F'$ (little competition) [56]; $\epsilon(F, s)$ is defined as the sum of Miyazawa-Jernigan interactions [62, 63] between residues $s_i$, $s_j$ in contact on structure $F$. The fitness of a protein $s$ (with respect to the native fold $F$) is defined as

$$\mathcal{H}(s|F) = -\log P_{nat}(s|F) \quad \text{with} \quad P_{nat}(F|s) = \frac{e^{-\epsilon(F,s)}}{\sum_{F'} e^{-\epsilon(F',s)}} \quad . \tag{13}$$

An MSA for the family $F$ can then be obtained by sampling from the effective Hamiltonian $\mathcal{H}$. To control for the mean Hamming distance from a given wildtype sequence $wt$ of the sampled MSA, we follow the procedure of [50] and sample from a biased Hamiltonian

$$\mathcal{H}^\gamma(s|F, wt) = \mathcal{H}(s|F) - \gamma\, d(wt, s) \quad . \tag{14}$$

As in [50], Monte Carlo sampling is performed with the Metropolis rule at effective temperature $\beta = 1000$ and with $T = 1000$ thermalization steps between each sampled sequence. Precise values of $D$ were obtained by sub-sampling and mixing four large alignments obtained with $\gamma = 0$, 0.025, 0.050, and 0.075. From each MSA, the computation of descriptors $\sigma^2$ and $D$ as well as training and performance assessment of Potts models, were performed as explained below for real proteins, with the difference that no re-weighting procedure was applied to lattice proteins data.

## Numerical estimation of bias and variance in Lattice Proteins

In the case of Lattice Proteins, we numerically computed the real fitness difference caused by single-point mutations as $\mathcal{E}_{ia} = \mathcal{H}(wt_{i \to a}) - \mathcal{H}(wt)$. For a given inferred Potts model, we then computed the bias $\bar{\mu}_{ia}^2$ and variance $\bar{\sigma}_{ia}^2$ of its delta-fitness predictors $\hat{\mathcal{E}}_{ia}$ as

$$\bar{\mu}_{ia}^2 = \left( \left[ \hat{\mathcal{E}}_{ia} - \mathcal{E}_{ia} \right] \right)^2 \tag{15}$$

$$\bar{\sigma}_{ia}^2 = \left[ \hat{\mathcal{E}}_{ia}^2 \right] - \left[ \hat{\mathcal{E}}_{ia} \right]^2 \quad , \tag{16}$$

where averages are computed over $n = 10$ inferences performed on as many sampled alignments with fixed number $B$ of sequences and mean Hamming distance $D$ to $wt$.

To relate these quantities to the single predictive performance value $\rho$ of the inferred model, we defined two global measures that account for all single-point mutations $(i, a)$:

$$\bar{\mu}^2 = [\bar{\mu}_{ia}^2]_{i,a} - [\bar{\mu}_{ia}]_{i,a}^2 \tag{17}$$

$$\bar{\sigma}^2 = [\bar{\sigma}_{ia}^2]_{i,a} \qquad , \tag{18}$$

where $[\cdot]_{i,a}$ denotes the averages over sites and mutations, and the global shift in the bias $[\bar{\mu}_{ia}]_{i,a}^2$ is removed, as the Spearman rank correlation $\rho$ is invariant under the addition of a constant to $\hat{\mathcal{E}}_{ia}$. In these numerical settings, some mutations are so deleterious that will never be observed in the data, and their effect is systematically estimated by regularization only. To avoid that these outliers dominate the averages above, we restricted our analysis to those mutations that satisfy an "observability" criterion of $|\mathcal{E}_{ia}| < \theta$. Unless specified differently, we use $\theta = 5.0$ throughout all Lattice Protein results.

## Discussion

In this work, we have investigated, through a combination of analytical and numerical approaches, how the nature (quantity, similarity to *wt*) of sequence data determine the capability of statistical models, with variable expressive power, to predict the fitness effects of single-point mutations. As expected from the bias-variance trade-off of statistics, simple models require few data to be inferred, but result in systematic prediction errors (bias $\mu$). Conversely, powerful models are in principle capable of expressing complex sequence-to-fitness relationships but their many defining parameters are subject to more statistical errors due to the limited amount of available data (variance $\sigma^2$). We have shown that a good predictor of performances was given by the sum $\mu^2 + \sigma^2$, and have analytically related the variance to the number of sequences $B$ in the alignment and the squared bias $\mu^2$ to the evolutionary depth, estimated through the mean Hamming distance $D$ to the mutated wild type sequence. Our theory was quantitatively confirmed by extensive tests on *in silico* lattice proteins for which the ground-truth fitness is known, and on mutagenesis datasets of 7 proteins families we have analyzed. Based on the results above, we then proposed a "focusing" procedure to optimally select the best subset of sequences from a multi-sequence alignment, and tested it on the 7 mutagenesis experiments. With this procedure, the least powerful, independent sites model, showed performances higher than fully connected graphical models trained on the same data for 4 out of 7 studied protein families, and comparable performances for the remaining ones.

An important finding of the present work is the so-called bias factor $J_0$, which relates the squared bias $\mu^2$ to the mean Hamming distance $D$ of the sequence data to the wild-type sequence: $\mu^2 \simeq J_0 D$. In our idealized theoretical framework, confirmed by simulations on *in silico* proteins, $J_0$ accounts for un-modelled epistasis, i.e., for the statistical properties of the fitness landscape that cannot be reproduced by the class of models considered. Though other sources of biases can be present in real data or in the inferred models due to regularization, and contribute to $J_0$, our result has two consequences, both conceptual and practical. First, it explicitly demonstrates that key information about the unmodeled features of fitness landscapes are, in principle, accessible even with models with limited complexity, constrained by data availability. From a practical point of view $J_0$ can be estimated through a regression of the performance $\rho$ vs. a linear combination of $D$ –chosen at will through subsampling of the multi-sequence alignment– and $\sigma^2$ –given by Eq (3)–, see Fig 5A; this procedure can therefore be applied to any protein family, for which sequence and mutagenesis data are available. Second, the meaning of $J_0$ emphasizes the role of the expressive power of the model in the relative

importance of the bias and variance terms, and to what extent each one of these factors affect performances. The value of $J_0$ is a good predictor of how much can be gained in performance by pruning the sequence data and focusing around the *wt* sequence (Fig 5E). This result entails that simpler models have higher potential for improvement in fitting a local neighborhood through focusing, and can overcome complex models when training data is appropriately selected.

Determining the optimal cutoff distance for focusing can theoretically be done following the quantity-relevance trade-off analysis presented above, *e.g.* using some already available mutagenesis experiment. We proposed an empirical rule that did not require any mutational information and was based on a signal-to-noise criterion. This empirical cutoff led to systematic improvement of performance for all tested families (S3 Fig).

Our focusing and modeling procedures could be further improved along several directions. First, in the the *K*-links Potts model considered here we have selected relevant links according to the Frobenius norms of the couplings of the inferred Potts model (equivalently, in the Direct Coupling Analysis, DCA). The rationale for this criterion is that the coupling norm is a good proxy for coevolution and contact between residues. Sparsity of the interaction graph can be enforced, within DCA, through $L_1$ regularization over the couplings [54, 61, 64]. However, in a related work [38], we have shown that DCA-based ranking is not an optimal predictor of relevance of couplings for protein function. Couplings can be better selected using a semi-supervised procedure, which exploits a subset of mutational data. Such optimally selected *K*-Links Potts models achieve a clear increase of the performances in predicting the effect of mutations.

Second, we have estimated, so far, the closeness of an alignment to the wild-type sequence through the average Hamming distance *D*. This choice is justified both by its simplicity, and the deep relation between *D* and the (squared) bias. However it would be worth considering more refined estimates for the distances, taking into account the phylogeny of the sequence data. Residue conservation can be assessed according to mutational history [30, 31], or to their relevance in the functionality under consideration. In addition, our focusing procedure could make use of alignment methods based on local homology, recently used to discover specific functionality proper to some protein subfamilies [65].

Our theoretical study could help improve models and alignment processing for predicting the effects of missense mutations and their impact in genetic diseases [6–10]. Natural alignments of sampled missense mutations are limited in depth and naturally focused around the human genome, making independent-site models (or *K*-link Potts models with small *K* values) especially adequate. It would be very interesting to apply our focusing approach to understand how to best select sequence data in this context.

The capability of deriving optimal independent-site models, whose parameters are tuned according to the region in the sequence space under focus, could be also be important for phylogeny studies. Inferring phylogenetic relations between a set of sequences requires the capability to compute transition probabilities under a mutation-selection process. Independent-site models are particularly attractive in this regard as they lead to mathematically tractable expressions for the transitions [66], but cannot describe complex sequence-to-fitness mappings. An alternative would be to use multiple focused independent-site models to compute transitions, adapted to the multiple portions of the sequence space explored by the phylogenetic tree.

Last of all, we stress that the question of how to select the best subset of data ensuring optimal performance given a statistical model is of interest in the field of proteins beyond fitness predictions, and, more generally, in machine learning. In the context of structural predictions, it is known that AlphaFold performances are sensitive to the input multi-sequence alignment; in CASP15 some methods found improved predictions by changing the way sequence data

were generated [67]. From a general machine-learning point of view, the present work bears some similarity with classical issues in statistics, in particular, the dependence of performance on the quantity of training data. In theoretical consistency frameworks, data are assumed to be generated independently at random from a fixed model distribution (sometimes referred to as the teacher), and then used to train another model with the same architecture (the student), see for instance [68–70] for applications to graphical model reconstruction. However, in our case, the teacher (fitness landscape) is of high and unknown complexity, while the student is much simpler (independent-site or sparse Potts models). Our goal was to provide theoretical support for a pruning strategy, in which data likely to be poorly modeled by the student are explicitly filtered out in the training phase. Our focusing procedure is, in this sense, conceptually related to local regression methods, such as moving least squares approaches, which aim at locally fitting a function from data. It is therefore expected that it will find applications beyond the prediction of fitness considered here. For instance, focused independent-site models could be useful in the context of gene expression, where microarray data generally suffer from missing values, impeding the use of many multivariate statistical methods [71].

## Supporting information

**S1 Text. Supplementary information.** Contains Appendices A and B.
(PDF)

**S1 Fig. Supplementary figure 1.** Same as Main text Fig 2D for all protein families except RNA-Bind (shown in Main text Fig 2D): systematic analysis of the predictive power $\rho$ as a function of the mean Hamming distance $D$ of sub-alignments with fixed size $B$ (left panels), and of the sub-alignment size $B$ at fixed Hamming distance $D$ (right panels). Each point represents the binned average and standard deviation of several sub-samples obtained at the corresponding values of $D$ and $B$ (see Methods). All significance levels refer to Spearman rank correlation. * $P < 0.05$; ** $P < 0.01$; *** $P < 0.001$.
(PDF)

**S2 Fig. Supplementary figure 2.** Same as Main text Fig 4A&4B for all protein families except RNA-Bind (shown in Main text Fig 4A&4B): predictive performance of single-point mutations using the independent-site models, as a function of the squared bias and variance estimated from the alignments, separately (left and right panels) and combined (central panel).
(PDF)

**S3 Fig. Supplementary figure 3.** Top: single mutation prediction performance of the independent Potts model ($K = 0$) along the focusing axis (as a function of the cutoff distance $D_0$) for the 7 studied protein families. Black dashed lines indicate the optimal cutoffs $d^{opt}$; blue lines indicate the predicted cutoffs $d^{bv}$ by minimizing the linear sum of bias and variance; the light blue lines indicate the predicted cutoff from the signal-to-noise heuristic $d^{snr}$. Green areas highlight the performance increase from the full alignment ($d_c = N$) to the predicted cutoff $d^{bv}$. Yellow areas indicate the remaining performance increase to the optimal cutoff $d^{opt}$. Horizontal dashed grey lines indicate the performance reported in [37] with a fully connected Potts model inferred by pseudo-likelihood. Bottom: distribution of the hamming distance to the wildtype $D$ of sequences in the MSA. Black dashed lines indicate the optimal cutoff at which the best performance is reached. $B_{\text{eff}}^{\text{opt}}$ is the effective number of sequences remaining in the MSA at the optimal cutoff. Refer to Table 1 in Methods for the original number of sequences in the MSA.
(PDF)

**S4 Fig. Supplementary figure 4. a**: Single mutation prediction performance of the independent model at the predicted optimal cutoff using the SNR method, as a function of the SNR threshold, for the 7 protein families. **b** comparison between performance without any cutoff (MSA full) and performance at the cutoff predicted by using the rule of thumb SNR = 3.
(PDF)

**S5 Fig. Supplementary figure 5.** Relation between the bias factor $J_0(K)$ and improvement $\Delta\rho$ ($d^{opt}$) for the optimal focusing cutoff for the 7 studied protein families. For each family, $K$ is varied between 0 and $N$ (number of sites in the alignment).
(PDF)

# Acknowledgments

The authors are grateful to J. Tubiana and M. Molari for insightful discussions, and to J. Fernandez de Cossio Diaz and E. Mauri for a careful reading of the manuscript.

# Author Contributions

**Conceptualization:** Lorenzo Posani, Rémi Monasson, Simona Cocco.

**Data curation:** Lorenzo Posani, Francesca Rizzato.

**Formal analysis:** Lorenzo Posani, Rémi Monasson, Simona Cocco.

**Funding acquisition:** Rémi Monasson, Simona Cocco.

**Investigation:** Lorenzo Posani, Rémi Monasson, Simona Cocco.

**Methodology:** Lorenzo Posani, Francesca Rizzato, Rémi Monasson, Simona Cocco.

**Software:** Lorenzo Posani, Francesca Rizzato.

**Supervision:** Rémi Monasson, Simona Cocco.

**Writing – original draft:** Lorenzo Posani, Rémi Monasson, Simona Cocco.

**Writing – review & editing:** Lorenzo Posani, Rémi Monasson, Simona Cocco.

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
