## [Decision Letter · Decision Letter 0]

6 May 2023

Dear Dr Cocco,

Thank you very much for submitting your manuscript "Infer global, predict local: quantity-quality trade-off in protein fitness predictions from sequence data." for consideration at PLOS Computational Biology.

As with all papers reviewed by the journal, your manuscript was reviewed by members of the editorial board and by several independent reviewers. In light of the reviews (below this email), we would like to invite the resubmission of a significantly-revised version that takes into account the reviewers' comments.

We cannot make any decision about publication until we have seen the revised manuscript and your response to the reviewers' comments. Your revised manuscript is also likely to be sent to reviewers for further evaluation.

Sincerely,

Rachel Kolodny

Academic Editor

PLOS Computational Biology

Arne Elofsson

Section Editor

PLOS Computational Biology

Reviewer's Responses to Questions

**Comments to the Authors:**

Reviewer #1: This is a well-written paper on the important topic of deciding which kind of data (large but of worse quality or of more moderate size but of better quality) is more appropriate to infer fitness landscapes. The inference procedure considered in the paper is model learning in exponential families (or Direct Coupling Analysis). As is well known today this is inferior to deep learning / AlphaFold / on the flagship application of predicting protein structure, but still a method of choice when large training data is not available. Also, this kind of inference procedure is, though also somewhat complex, still sufficiently simple that it may be possible to analyse performance ab initio.

Several kinds of data are considered in the paper, and are clearly described. The results are also presented clearly. A strong aspect of the paper is the extensive use of data from deep mutational scans.

I hence recommend acceptance, provided some minor adjustments are made.

1. The authors focus on a mean-square characterization of errors, but as the authors surely know this is not the only possibility. It is also not the main characterization of performance of this class of methods on real data; instead a version of "top-k" is more commonly adopted.

This importance of inference criteria in this context has been stressed e.g. by Aurell and co-workers (most recently in arXiv:2205.00750), but other points of view can also be found in the literature.

Important technical contributions include

M. Vuffray, S. Misra, A. Lokhov, and M. Chertkov. Interaction screening: Efficient and sample-optimal learning

of Ising models. In D. D. Lee, M. Sugiyama, U. V. Luxburg, I. Guyon, and R. Garnett, editors, Advances in

Neural Information Processing Systems 29, pages 2595–2603. Curran Associates, Inc., 2016.

A. Y. Lokhov, M. Vuffray, S. Misra, and M. Chertkov. Optimal structure and parameter learning of Ising

models. Sci. Adv., 4(3):e1700791, March 2018.

J. Berg. Statistical mechanics of the inverse Ising problem and the optimal objective function. J. Stat. Mech:

Theory Exp., 2017(8):083402, 2017.

mostly for other variants of DCA than considered by the authors, but also including pseudo-likelihood maximization. The gist of these contributions is that if the goal is "top-k" (as it has often been in practice), then very few samples are often enough to achieve good performance.

I ask the authors to include in a revised version of their paper a short section where they conceptualize this issue and discuss various pros and cons.

2. Two relevant theoretical papers which also deserve to be cited and briefly commented are

L. Bachschmid-Romano and M. Opper. A statistical physics approach to learning curves for the inverse ising

problem. Journal of Statistical Mechanics: Theory and Experiment, 2017(6):063406, jun 2017.

A. Abbara, Y. Kabashima, T. Obuchi, and Y. Xu. Learning performance in inverse ising problems with sparse

teacher couplings. Journal of Statistical Mechanics: Theory and Experiment, 2020(7):073402, jul 2020.

3. The authors are encouraged to survey the theoretical literature to make sure that later paper in the direction of the above are not missed. It is important for coherence of the field that contacts between theory-driven and data-driven contributions are maintained.

Reviewer #2: Uploaded as attachment

**Have the authors made all data and (if applicable) computational code underlying the findings in their manuscript fully available?**

Reviewer #1: Yes

Reviewer #2: **No: **I could not find a link to the code.

PLOS authors have the option to publish the peer review history of their article (what does this mean?). If published, this will include your full peer review and any attached files.

Reviewer #1: No

Reviewer #2: No
---

## [Decision Letter · Decision Letter 1]

15 Sep 2023

Dear Dr Cocco,

We are pleased to inform you that your manuscript 'Infer global, predict local: quantity-relevance trade-off in protein fitness predictions from sequence data' has been provisionally accepted for publication in PLOS Computational Biology.

Best regards,

Rachel Kolodny

Academic Editor

PLOS Computational Biology

Arne Elofsson

Section Editor

PLOS Computational Biology

Reviewer's Responses to Questions

**Comments to the Authors:**

Reviewer #1: I am satisfied with the answer of the authors, and recommend acceptance.

Reviewer #2: Thank you for thoroughly addressing my comments.

**Have the authors made all data and (if applicable) computational code underlying the findings in their manuscript fully available?**

Reviewer #1: Yes

Reviewer #2: None

PLOS authors have the option to publish the peer review history of their article (what does this mean?). If published, this will include your full peer review and any attached files.

Reviewer #1: No

Reviewer #2: No

---

## [Editor Report · Acceptance letter]

17 Oct 2023

PCOMPBIOL-D-23-00174R1 

Infer global, predict local: quantity-relevance trade-off in protein fitness predictions from sequence data

Dear Dr Cocco,

I am pleased to inform you that your manuscript has been formally accepted for publication in PLOS Computational Biology. Your manuscript is now with our production department and you will be notified of the publication date in due course.

With kind regards,

Zsofi Zombor
